# Clinical Outcomes of 3D-Printed Titanium Patient-Specific Implants in Lumbar Interbody Fusion: A Prospective Clinical Trial with a Systematic Review of Conventional Techniques

**DOI:** 10.3390/jpm15070320

**Published:** 2025-07-16

**Authors:** Kevin A. Seex, Ralph J. Mobbs, Marc Coughlan, Matthew Pelletier, William R. Walsh, Jackson C. Hill, William C. H. Parr

**Affiliations:** 1Macquarie Neurosurgery, Macquarie University Hospital, Macquarie Park, NSW 2109, Australia; 2Neuro Spine Clinic, Prince of Wales Private Hospital, Randwick, NSW 2031, Australia; 3NeuroSpine Surgery Research Group (NSURG), Randwick, NSW 2031, Australia; 4Coastal Neurosurgery, Gosford Private Hospital, Gosford, NSW 2250, Australia; 5Surgical and Orthopaedic Research Laboratories (SORL), Prince of Wales Clinical School, Faculty of Medicine, University of New South Wales (UNSW), Randwick, NSW 2031, Australia; 63DMorphic Pty Ltd., Matraville, NSW 2036, Australia; jackhill2412@gmail.com

**Keywords:** spinal fusion, patient-specific implant, interbody cage, Lumbar Interbody Fusion, case series, patient-reported outcomes

## Abstract

**Background**: Lumbar Interbody Fusion (LIF) is a surgical procedure aimed at addressing a range of pathological conditions affecting the structure and function of the spine. Patient-Specific Interbody Cages (PSICs) are an emerging technology that are used in LIF; however, there is a lack of clinical outcome data, making it difficult to assess the potential risks, benefits, and value of PSICs. The purpose of this present study is to contribute data to the field on this new emerging technology. The aims were to investigate Quality of Life (QoL), pain, and the complications of PSICs in LIF. To provide a comparative cohort, we performed a systematic review of patient-reported outcomes of conventional fusion techniques. **Methods**: This study reports on a multi-surgeon, multi-centre clinical trial of patients with lumbar degenerative disc disease, necessitating discectomy and fusion. All patients underwent LIF procedures with 3D-printed PSICs. Pain Visual Analogue Scale (VAS) and QoL (EQ-5D) scores were collected pre-operatively and at 6m, 12m, and 24m post-operatively. For comparative purposes, we performed a systematic review of the VAS scores from traditional LIF cages and analysed the Australian Spine Registry QoL data. **Results**: The literature search yielded 4272 publications. The studies were subdivided into four groups based on the interbody device type. All the groups demonstrated improvements in the VAS (for back pain) scores post-operatively. In total, 78 patients (109 instrumented levels) underwent LIF procedures with 3DP PSICs. There were three reoperations (3.8%) and no revisions of any PSIC. The mean VAS scores improved significantly (*p* < 0.01) from 7.85 (1.50 SD) pre-operatively to 2.03 (2.13 SD) at 24 months post-operatively. The mean QoL index scores improved significantly (*p* < 0.01) from a pre-operative 0.257 (0.332 SD) to 0.815 (0.208 SD) at 24 months. **Conclusions**: The systematic review indicated that device fixation to the interbody space was associated with lower VAS scores. The results from the investigational cohort suggest that PSICs may represent a new progression in implant design for spinal fusion, with an associated clinical benefit for LIF.

## 1. Introduction

Lumbar Interbody Fusion (LIF) is a common surgical procedure aimed at addressing a range of pathological conditions affecting the structure and function of the spine [1]. Although several different approaches are utilized for LIF, the fundamental principles remain consistent. The key surgical steps involve the retraction, discectomy, endplate preparation, and implantation of an interbody fusion cage. The short-term objectives of LIF are to alleviate pressure on the neural structures by distracting the intervertebral space and to stabilise the joint to reduce the dynamic components that cause clinical symptoms [2]. In the mid- to longer term, the objective is to promote a fast and controlled fusion between the adjacent vertebrae. It is important that fusion occurs with a balanced alignment, as spinal alignment has been linked to patient-reported outcomes [3]. Spinal alignment can be affected by the cage placement in the interbody space, as well as by the design (e.g., the lordotic angle) of the interbody cage [4].

Following the inception of interbody cage use with the “Bagby Basket” in the 1980s, cages have undergone design and construction changes in terms of materials, dimensions, and fixative methods [5]. There have been significant developments in LIF cages, from autograft blocks and Bagky and Kuslich (BAK) cages to modern cages with and without integral screw fixation and now to patient-specific implants. Cage improvements have also been supplemented by advances in bone grafting and the osteobiologics that are packed in and/or around the cages [6]. However, issues remain with reportedly high reoperation rates (18–23%) [7], pseudoarthrosis, adjacent segment disease, and cage subsidence [8].

Lumbar interbody spaces are neither symmetrical nor the same size and shape across the lumbar spine [9]. Standard Off-The-Shelf (OTS) devices provide limited options to accommodate the unique anatomical features and sizes. Consequently, surgeons may need to reshape the vertebral endplates to fit the symmetrical, flat-sided devices using techniques such as bone burring and rasping. This can remove the cortical layer of bone on the endplate and expose the weaker cancellous bone underneath [10].

Three-dimensional printing (3DP), also known as additive manufacturing and rapid prototyping, was developed in 1984 (the first patent was in 1986 [11]) and is widely perceived as a technological advancement in manufacturing capability. The underlying principle of 3DP is the layer-by-layer addition of material to build a 3D structure. Practically, this involves generating 3D computer models, slicing them into 2D cross-sections and physically producing these by printing each layer [12]. Three-dimensional printing allows for previously un-manufacturable designs to be realised and has become cheaper, faster, and easier to use, resulting in widespread application, such as in healthcare, where research spans almost every major medical specialty [12,13]. More recently, 3DP in biocompatible materials has been applied to spinal surgery to produce anatomical models, screw guides, and implants. Models of patient spines have been used for patient education, training, and planning, both pre-operatively and intraoperatively [14,15]. Reported spinal patient-specific implants include sacral prostheses [16], corpectomy cages [17], and interbody cages [18] at the cervical, thoracic, and lumbar levels [19]. Custom surgical instruments include protective jigs [14], pedicle screw guides [20], and osteotomy guides [21,22].

Three-dimensional-printed Patient-Specific Interbody Cages (PSICs) are designed to fit the interbody space and provide the desired sagittal and coronal alignment corrections [23]. This approach offers an alternative to standard OTS devices, but no studies have identified statistically significant differences in patient-reported outcomes for patients treated with patient-specific devices. This may be due to underpowered study design, making it difficult to assess the potential value, benefits, and risks of PSICs. The purpose of this present study is to contribute data to the field on this new emerging technology of PSICs.

The aims were to investigate health-related Quality of Life (QoL), pain, and complications, including revision/reoperation rates, to assess the potential benefits and risks of PSICs. To provide a comparative cohort, we also present a systematic review on the patient-reported outcomes of traditional fusion techniques, along with an analysis of the Australian Spine Registry (ASR) [24].

The null hypothesis for the statistical testing was that there is no difference between the PSIC and OTS device clinical scores at any time point.

## 2. Materials and Methods

### 2.1. PSIC Cohort

#### 2.1.1. Study Design

This study was designed as a single-region (country), multi-centre, multi-surgeon prospective clinical trial of patients undergoing LIF using 3DP PSICs.

#### 2.1.2. Ethical Approval

Ethics approval was obtained from the South Eastern Sydney Local Health District (project identifier 2021/ETH11998) on 6 February 2022, and the trial was registered with the Australian Therapeutic Goods Administration (CT-2022-CTN-00042-1) on 19 February 2022. Informed consent was obtained from all the subjects involved in this study. The patients were recruited from 19 February 2022 to 15 July 2024.

#### 2.1.3. Sample Size and Power Analysis

The sample size required to detect a clinically meaningful difference for this study was calculated using the following formula [25]:$$N=\frac{\sigma^{2}{\left(z_{1-\beta }+z_{1-\alpha /2}\right)}^{2}}{{\left(\mu_{0}-\mu_{1}\right)}^{2}},$$ where *N* is the number of participants, σ is the known SD, $\mu_{0}$ is the known mean, and $\mu_{1}$ is the anticipated mean of the investigational group. The risk of a type I error was set to 5% (*z*_1−*β*_ = 0.84), and the risk of a type II error was set to 20% (*z*_1−*α*_/2 = 1.96) [25]. Gornet et al. reported a 4.9 (3.1 SD) improvement in the VAS scores for spinal fusion patients 24 months post-operatively. An improvement of 1.4 in the VAS scores would be considered clinically significant [26].


$$N=\frac{\left(3.1^{2}{\left(0.84+1.96\right)}^{2}\right)}{{\left(4.9-6.3\right)}^{2}}=38.$$


#### 2.1.4. Inclusion and Exclusion Criteria

All patients were selected for surgery due to the discogenic and/or mechanical low back pain symptomatic of degenerative disc disease (DDD). Skeletally mature (>18 years) patients were included. Worker’s compensation patients were excluded from the cohort. Previous studies have indicated that factors related to worker’s compensation can influence the reporting of the Visual Analogue Scale (VAS) and QoL scores [27,28]. 

#### 2.1.5. PSIC Design

3DMorphic Pty Ltd. (Sydney, Australia) performed Virtual Surgery Planning (VSP) [23]; 3DP biomodelling; and PSIC design, manufacture, and distribution (Figure 1 and Figure 2). The PSICs were designed according to the processes described previously [17]. Before surgery, the patients underwent Computed Tomography (CT) scanning. These scans were used to reconstruct 3D models of the pathological anatomy with Materialise MIMICS and 3Matic (Leuven, Belgium) [29].

Pre-operatively, the surgical goals were established through the VSP in consultation with the surgical team. These goals included distraction, the restoration of sagittal balance, and the relief of clinical symptoms.

The Ti-6Al-4V (Ti64) (Biomedical Grade 5) PSICs were 3D-printed using Direct Metal Laser Solidification. Ti64 was used due to its biocompatibility, relatively low stiffness (~110 GPa), and lower propensity to create radiographic artefacts compared with steel (316l) or Cobalt Chrome metallic alternatives [30]. Ti64 is a well-established 3D-printable material for implantable medical devices. While bone apposition is influenced by several factors, Ti64 often exhibits superior bone contact compared to smooth Polyetheretherketone (PEEK), as surface roughness is often a key factor in apposition [31,32,33]. Although the bone-contacting properties of PEEK can be improved with coatings, bulk additions, and macro features, Ti64 remains a frequently used material in load-bearing applications [34,35,36].

#### 2.1.6. Operative Technique

The patients underwent anterior column LIF with some cases supplemented by posterior rods and screws. Anterior (ALIF) and lateral (ATP/LLIF/OLIF) approaches were utilized in this study, both of which followed a similar operative technique once at the stage of discectomy and the implantation of the device. For the ALIF approach, the patient was positioned in the supine position, and an incision was made to expose the retroperitoneum. For lateral cages, with the patient in a lateral position, an oblique approach through the retroperitoneum was used, accessing the disc space and retracting the psoas without neuromonitoring using Anterior to Psoas (ATP) instrumentation (Relax Retractors, Sydney, Australia). Image intensification was used to confirm the pathological levels, followed by a discectomy that involved careful endplate preparation to avoid damaging the endplate. The cages were then inserted, and fluoroscopy was used to verify their alignment. Once the surgeon had chosen the cage size, the graft windows were packed with bone graft substitutes according to surgeon preference and hospital availability. These included allografts (Australian Biotechnologies Pty Ltd., Sydney, Australia), autografts, and a bioactive glass (GlassBone™, Noraker, Lyon-Villeurbanne, France). Rh-BMP2 and similar bone morphogenic protein products were not used. The cages were then secured via integral screw fixation. Depending on the patients’ anatomy and surgeon preference, ALIF cages had options of three or four screws, and LLIF cages were secured with either one or two screws.

#### 2.1.7. Post-Operative Care

After the surgery, the patients were transferred to the Intensive Care Unit, and their lower limbs were monitored for neurological issues. They were given analgesics, prophylactic antibiotics, and venous thromboembolic prophylaxis. The patients were allowed to sit upright to 30 degrees from the horizontal position. During recovery, the patients were educated on spinal precautions and advised on “red flag” symptomatology. The patients were braced with a lumbar corset to serve as a psychological reminder to maintain a neutral lumbar spine position and avoid bending, lifting, and twisting for six weeks.

#### 2.1.8. Outcome Measures

The pre-operative baseline and post-operative patient outcomes were assessed using VAS and EQ-5D-5L scores [37]. QoL data was determined using the EQ-5D-5L questionnaire and health state values (see Appendix A) [38,39,40,41]. QoL indices measure health on a scale from 1 (full health) to 0 or less (death-equivalent). These indices provide the QoL input to calculate Quality Adjusted Life Years (QALYs) and are widely used in cost–utility analysis to provide an economic evaluation of health technologies [39]. These two surveys, along with conversations about the patient’s recovery, provided a comprehensive range of quantitative and qualitative information.

#### 2.1.9. Follow-Up Protocol

The patients were followed up at regular intervals by their surgeon’s rooms, and PROMs were collected at 6-month, 12-month, and 24-month intervals.

To provide a comparative cohort, we analysed the results from the 2023 ASR. This is a temporally relevant cohort and includes results from a similar (Australian) population. As the ASR does not report on VAS scores, we undertook a systematic review of the literature to develop a comparative cohort for this metric. 

### 2.2. Systematic Review of OTS Comparator Group

#### 2.2.1. Search Strategy

We performed a literature search following the preferred reporting items for systematic reviews and meta-analyses (PRISMA) guidelines [42]. The systematic review was registered with PROSPERO (CRD420251061430). The search was conducted on 15 May 2024, using PubMED, Embase, and the Cochrane Central Register of Controlled Trials (CENTRAL). The search strategy utilised MeSH headings and the search terms included (“clinical outcomes” OR “Patient Reported Outcome Measures” OR “Patient Reported Outcomes”) AND “lumbar” AND (“anterior” OR “lateral” OR “extreme” OR “oblique”) AND (“spinal fusion” OR “arthrodesis” OR “interbody fusion”).

#### 2.2.2. Systematic Review Inclusion and Exclusion Criteria

We considered Randomised Controlled Trials (RCTs) and cohort studies for this analysis. We reviewed studies published as full texts and did not limit inclusion by date or language of publication. We considered patients who were treated for conditions indicated for use with 3DMorphic’s PSICs. Studies were included if patients were skeletally mature (≥18 years), with at least one of the following symptoms: (i) DDD resulting in neural dynamic compression due to loss of disc functionality, (ii) spondylolisthesis/retrolisthesis, (iii) spinal stenosis, (iv) foraminal stenosis, and (v) spinal deformity.

We included the following surgical approaches: ALIF, X/LLIF, and OLIF. These may also have been supplemented with posterior fixation (circumferential fusion). Studies that only investigated PLIF and/or TLIF were excluded.

#### 2.2.3. Systematic Review Outcome Measures

We searched for studies that reported on PROMs with long-term follow-up (defined as ≥12 months) that could be compared to the PSIC cohort. Studies with early follow-up (<12 months) were excluded. The key outcome measure was VAS (back pain).

#### 2.2.4. Risk of Bias Assessment

We used the criteria outlined in the Cochrane Handbook for Systematic Reviews of Interventions to assess the risk of bias for each study [43]. The risk of bias was assessed according to the following domains: random sequence generation (selection bias); allocation concealment (selection bias); the blinding of participants and personnel (performance bias); the blinding of outcome assessment (detection bias); incomplete outcome data (attrition bias); selective reporting (reporting bias); and any other bias (e.g., financial or other conflicts of interest).

#### 2.2.5. Data Extraction and Analysis

All the information was collected and analysed independently by two authors (J.H. and W.P.). The study characteristics and reported results were extracted and tabulated using Microsoft Excel. The study design, patient populations, intervention methods, comparator groups, and outcome measures were assessed against previously described eligibility criteria to identify the studies to be synthesised. The study authors were contacted in cases where there were missing or incomplete reported summary statistics (e.g., a missing SD). We extracted the intervention type, cage information, comparator groups, sample sizes, means, and SD of the VAS scores at each of the available follow-ups from the final chosen studies.

Consistent with Vali et al. [44], the studies were weighted based on their sample size. This method reflects the relative contribution of each study to the total estimated effect (see Appendix A).

### 2.3. Statistical Analyses

A statistical analysis was performed to evaluate the clinical outcomes. Two-tailed student T-tests were performed between the pre- and post-op scores, and between the different patient groups at each study time point. Two-tailed tests were appropriate as the null hypothesis was no difference between the PSIC and OTS groups.

The threshold for statistical significance was set at *p* < 0.05. All the statistical and regression analyses were conducted using the Python programming language (Python 3.11.3) with the scipy.stats statsmodels modules [45].

## 3. Results

Pre-operative and at least one post-op time point clinical scores were collected for 78 patients (109 operative levels) who were treated via LIF with 3DP PSICs. Individual level patient demographics are presented in Appendix A Appendix A.

### 3.1. Systematic Review

The literature search yielded 4272 publications. After duplicates were removed, 4064 records were screened. In total, 121 full-text articles describing 104 different studies were retrieved. There were 14 studies that met the inclusion criteria and were included in the qualitative and quantitative synthesis [46,47,48,49,50,51,52,53,54,55,56,57,58,59]. A flow diagram of the selection process is outlined in Figure 3.

Across the 14 RCTs, there were results for 860 patients treated with a comparable fusion technique. These were categorised into patients who received Femoral Ring Allografts (FRAs) (274 patients), BAK style cages (344 patients), and non-integral screw fixation (NISF) cages (242 patients). There were insufficient data from the 14 RCTs to create a meaningful integral screw fixation (ISF) cage group, so cohort studies were included (508 patients) [27,53,60,61,62,63,64,65,66]. The median study date for each group was 2009, 2011, 2017, and 2019 for the FRAs, BAK and integral fixation cages and non-integral fixation cages, respectively. Diagrammatic illustrations of the devices used in these groups are shown in Figure 4. The patient demographics are presented in Table 1. The risk of bias table from the 14 RCTs is presented in Table 2. The study characteristics are outlined in Table 3, and the raw results are summarised in Table 4.

### 3.2. Patient-Reported Outcome Measures (PROMs)

In the investigational cohort, there were 78 patients with pre-operative and at least one post-operative follow-up time point. Post-operative data was collected for 50 (VAS) and 44 (QoL) at 6 months; 39 (VAS) and 38 (QoL) at 12 months; and 44 (VAS) and 44 (QoL) at 24 months follow-up time points. The patients were not always available for data collection at all time points. To contribute the most comprehensive data possible, patients with incomplete follow-up were also included.

#### 3.2.1. VAS Clinical Outcomes

All the patient cohorts demonstrated statistically significant improvement in VAS scores (all post-op time points compared to the pre-operative baseline). The PSIC cohort had significantly greater improvement at 24 months compared to all the comparator groups (Table 5 and Figure 5). The final VAS scores for the PSIC patients at 24 months were also significantly less than all comparator groups (*p* < 0.05).

#### 3.2.2. Quality of Life (QoL) Clinical Outcomes

We used results from the 2023 ASR [24,40] to compare the QoL data. The mean QoL index results for the PSIC cohort improved significantly from 0.257 (SD 0.332) at the pre-operative baseline to 0.815 (SD 0.208) at 24 months post-operatively (see Table 6 and Figure 6). This can be compared to an improvement of 0.275 for the ASR patients.

The mean pre-operative QoL index was lower in the PSIC cohort than in the ASR patients. Considering this, we stratified the results based on a patient’s pre-operative QoL score. The groups were patients with pre-op QoL scores less than 0.2, between 0.2 and 0.5, and greater than 0.5 (see Figure 7).

### 3.3. Revision and Reoperation Results

We investigated the revision and reoperation rates in the PSIC cohort. Revision was defined as all or part of the original PSIC configuration being changed or removed [67]. Reoperation was defined as any surgical operation to the instrumented or adjacent levels after surgery.

In the investigational cohort, there were three reoperations (3.8%) and no revisions of any PSIC. Of the three reoperations, two patients had posterior decompression and fusion, and one patient required revision of the posterior instrumentation due to broken rods. All three reoperations occurred within 12 months of the initial surgery.

## 4. Discussion

This present study sought to provide clinical outcome data for a statistically significant cohort of patients treated with a PSIC to assess the potential benefits and risks of PSICs. The systematic review of conventional fusion techniques provides a comparison to the PSIC cohort. The patients in the PSIC cohort had on average 1.4 levels treated, which was similar across the OTS groups (with a mean of 1.3). In total, 59% of the PSIC patients had supplemental posterior instrumentation, compared with 0% of BAK, 24% of ISF, 95% of NISF, and 100% of FRA patients. Avoiding supplemental fixation in some patients may help reduce the risks associated with the additional posterior instrumentation [68]. On the other hand, surgeons are increasingly focused on restoring sagittal balance through a circumferential technique [69].

The results of this present study refute the null hypothesis: the PSIC cohort demonstrated statistically better VAS scores 24 months post-operatively compared with all other device groups. The QoL index scores and improvements were also greater than the ASR patients 24 months post-operatively. As a result of the missing and/or incomplete data for some patients, we have not performed repeated measures analyses. A regression analysis was performed to test for significance between patient demographics and outcomes. A one-year increase in age was associated with a 1.37% decrease in post-op VAS scores (*p* < 0.05). Age was not a significant predictor for QoL scores. Other independent variables including gender, number of levels treated, treated level, and surgical approach did not demonstrate statistically significant associations with either outcome (*p* > 0.05 for all).

The systematic review indicated a trend of reduced post-operative VAS scores with advancements in fusion device design. Fixation to the interbody space, either via screws through the cage or BAK cages, was associated with lower VAS scores, indicating that anterior stabilisation influences clinical outcomes.

A systematic review by Giang et al. 2017 [8] presented a mean post-op VAS score of 2.9 for a stand-alone ALIF. This is comparable to the ISF and BAK cohorts in this present study, with mean 24-month post-op VAS scores of 2.93 and 3.03, respectively.

Although RCTs were used to generate the FRA, BAK, and NISF cohorts, there were insufficient data to create a meaningful ISF cohort. This meant that the ISF group included cohort studies, with the potential for a higher degree of bias. Most RCTs (10/14) failed to satisfy the blinding criteria; however, blinding patients and surgeons in device trials is difficult [70].

### 4.1. VAS

A VAS score under 3 is considered mild pain, with 3–7 being moderate pain and 7–10 severe pain [26]. An improvement of 1.4 in a VAS score is clinically significant [26]. Both the investigational cohort and integral screw fixation cages reduced the VAS scores to below 3, within the threshold of mild pain. The difference in the 24-month post-operative scores between the PSIC cohort and the integral fixation cages (Table 5 and Figure 5) was statistically significant (*p* < 0.05).

All groups showed a significant improvement in VAS scores 6 months post-operatively (Table 5) compared to pre-operative levels. This may be attributed to the distraction (the decompression of neural elements) and stabilisation achieved upon the implantation of the device [71].

Devices that are fixed to the bone, either with bone screws (the PSIC and ISF groups) or BAK cage designs, provide better immediate stabilisation than non-fixated, or free-floating interbody spacers such as FRAs [72,73]. As the joint is stabilised, there is a reduction in the dynamic components contributing to the patient’s pain. These groups (PSIC, ISF, and BAK cages) resulted in significantly lower mean VAS scores at both 6 and 24 months post-operatively when compared to the FRA group (*p* < 0.05). The ISF and BAK groups had lower rates of supplemental posterior fixation compared to the other groups. The NISF and FRA groups had the highest rates circumferential fusion and were associated with higher post-operative VAS scores. Higher post-operative VAS scores associated with circumferential fusion may reflect an increase in surgical trauma associated with a two-stage front and back circumferential fusion. These results indicate that screw fixation (BAK or ISF) of the interbody cage is a significant contributing factor to lower post-operative VAS scores.

### 4.2. QoL

The mean QoL index improved from 0.257 pre-operatively to 0.815 at 24 months for the PSIC patients. This can be compared to an improvement from 0.472 to 0.747 at 24 months in the ASR.

A QoL index is a measure of an individual’s health status that ranges from 0 or less (representing a health state equivalent to death) to 1 (representing full health).

### 4.3. Revision and Reoperation

In Australia, Lewin et al. [7] report an 18–23% reoperation rate within 2 years of an initial spinal fusion surgery. They also reported (in 2021, based on data between the fiscal years 2010 and 2018) that the cost of an episode of lumbar fusion was AUD 52,379 (AUD 63,867 in FY2023/24), with a cumulative cost of AUD 81,297 at 24 months (AUD 99,127 in FY2023/24) due to follow-up costs such as medical imaging and physiotherapy. This demonstrates the need to improve spinal fusion outcomes, so costly revision surgeries are avoided.

In the US, a technical report prepared for the US Centres for Medicare & Medicaid Services on readmission following spinal fusion surgery was 9% within 30 days for patients diagnosed with DDD [74].

Within the PSIC cohort in this present study, there was a 3.8% (three patients) readmission rate over the entire follow-up period. All three patients were readmitted beyond 30 days of the initial procedure, and no patients were readmitted within 30 days. One of the three patients was readmitted within 90 days.

### 4.4. Efficacy

A recent US study [75] using PSICs for complex adult spinal deformity demonstrated a better immediate achievement of the alignment goals compared to stock devices but did not report clinical results. Another study [76] also reported a 0% subsidence rate at 1 year post-operatively for ALIF PSICs, defined as grade 0 subsidence (0% to 24% loss of post-operative disc height). This is compared with other literature results reporting ALIF subsidence rates between 6% and 23.1% [76].

The data presented here does not allow a determination of why PSICs resulted in better clinical outcomes for patients, but these better outcomes may be attributable to the improved vertebral alignment correction with PSICs [3,75]; the stability of the fixed constructs creating a biomechanical environment conducive to long-term bone fusion [8,59]; and the proximity of the PSIC surfaces to adjacent bone, enabling early osseointegration and reducing subsidence risk ([10,76,77] and Figure 8).

However, further radiographic assessment of the fusion status would need to be performed to confirm this.

### 4.5. Limitations

This present study, though prospective, was not a randomised trial with a direct comparison between patients treated with PSIC and OTS devices. The PSIC cohort was restricted to Australian patients, whereas the comparative OTS device cohorts included patients from other, non-Australian, populations. Systematic reviews are at the top of the hierarchy of evidence, and an ideal study would have compared the OTS devices to PSICs with a systematic review of the literature. However, it was not possible to include PSICs in the systematic review because there is a lack of published clinical outcome data for these devices.

The ISF group in the systematic review was primarily comprised of cohort studies as opposed to RCTs. Most of the RCTs in the systematic review failed to satisfy the blinding criteria.

This study was funded by the BioMedTech Horizons program delivered by MTPConnect to 3DMorphic Pty Ltd., who were responsible for the device design and manufacture, leading to potential conflicts of interest and bias. The potential direction of bias would be in favour of superior clinical outcomes for the PSIC patients. 3DMorphic Pty Ltd. was not involved in any patient selection, and the company did not refuse to make any PSIC for any patients when requested by the surgeon. Data imprecision is relatively low, as indicated by the low SD for the QoL and VAS scores for the PSIC patients.

The results presented in this present study are for PSICs produced by a single company. Therefore, the results presented here may not be applicable to PSICs produced by other companies using different design and manufacturing methods. The utilisation of patient-specific implants is increasing in clinical practice. There is, however, a notable paucity of literature regarding clinical outcomes for patients treated with PSIC devices.

This present study seeks to address this gap by objectively evaluating PSIC outcomes, acknowledging the potential limitations inherent in the study design. The data collected by PSIC manufacturers as part of clinical studies and/or regulatory obligations for post-market surveillance allows the scientific community to make informed, data-driven decisions concerning patient treatment options.

## 5. Conclusions

The results in this present study demonstrate that PSICs are a safe and effective treatment for intractable mechanical low back pain and/or radiculopathy, significantly reducing pain and improving QoL.

The systematic review showed that advancements in fusion devices have led to a steady improvement in clinical outcomes. Devices with additional screw fixation reported superior clinical outcomes in terms of back pain reduction.

The results suggest that PSICs may represent the next improvement step in the treatment of spinal fusion patients, with reduced back pain, increased QoL, and greater value for patients.

These results provide insight into the relationship between clinical outcome and device design. The evidence indicates that there is a clinical benefit associated with PSICs for LIF. Future work should evaluate clinical outcomes with PSIC use in cervical, corpectomy, and arthroplasty operations.

## Figures and Tables

**Figure 1 jpm-15-00320-f001:**
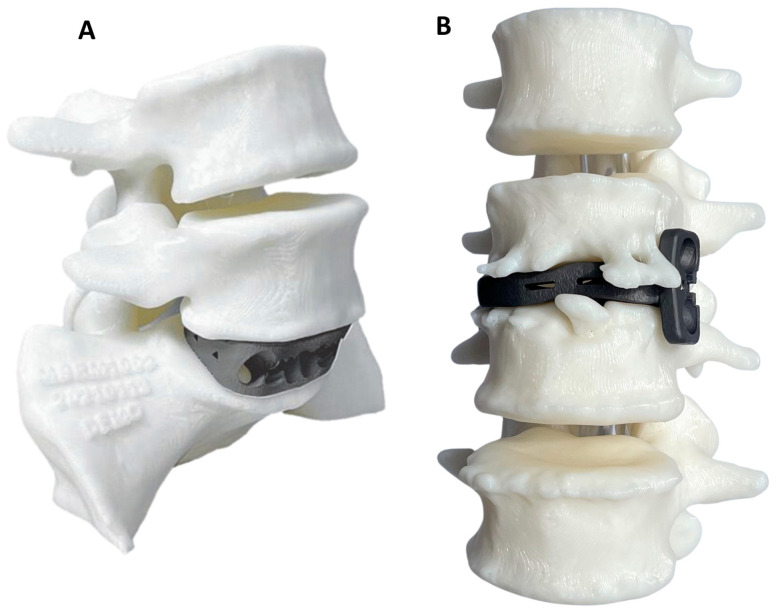
Examples of anatomical biomodels with patient-specific (**A**) ALIF and (**B**) LLIF cages.

**Figure 2 jpm-15-00320-f002:**
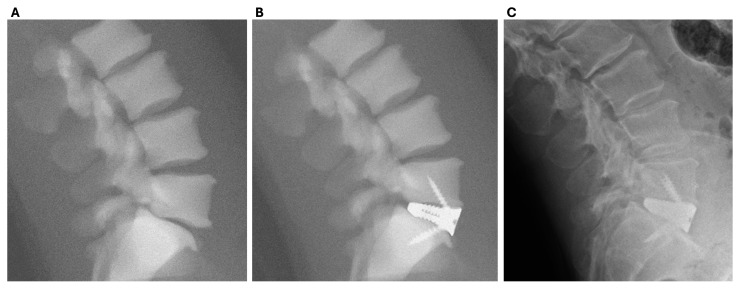
Simulated (**A**,**B**) and actual (**C**) X-ray images as a part of VSP demonstrating (**A**) pre-operative anatomy; (**B**) planned post-operative result produced through VSP; (**C**) actual achieved 2-month post-operative result.

**Figure 3 jpm-15-00320-f003:**
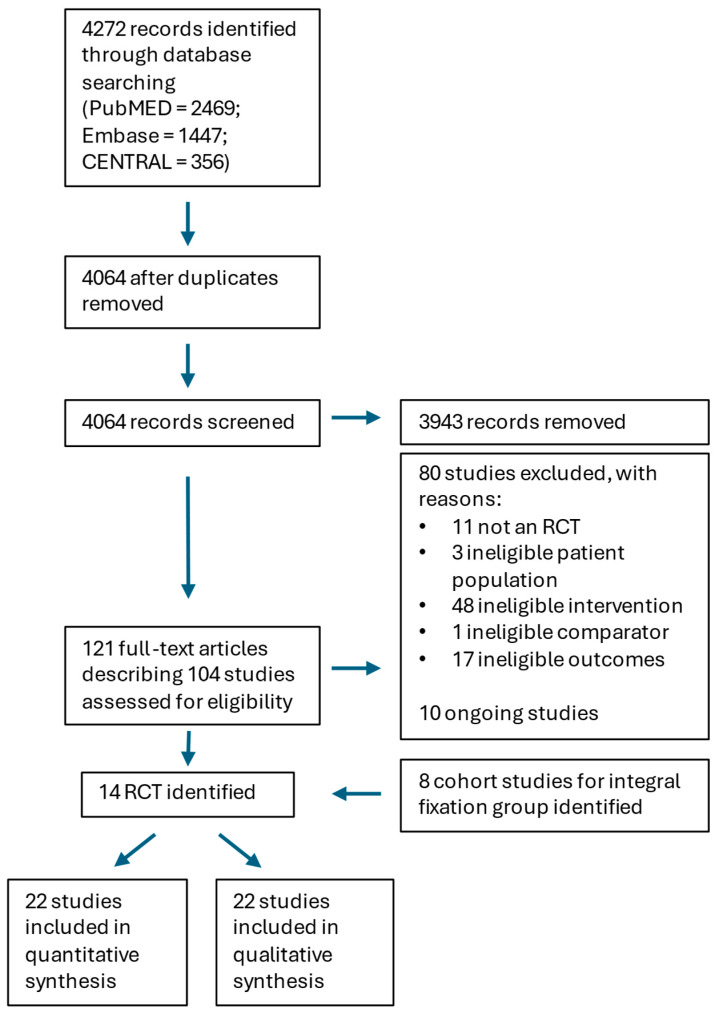
PRISMA study flow diagram.

**Figure 4 jpm-15-00320-f004:**
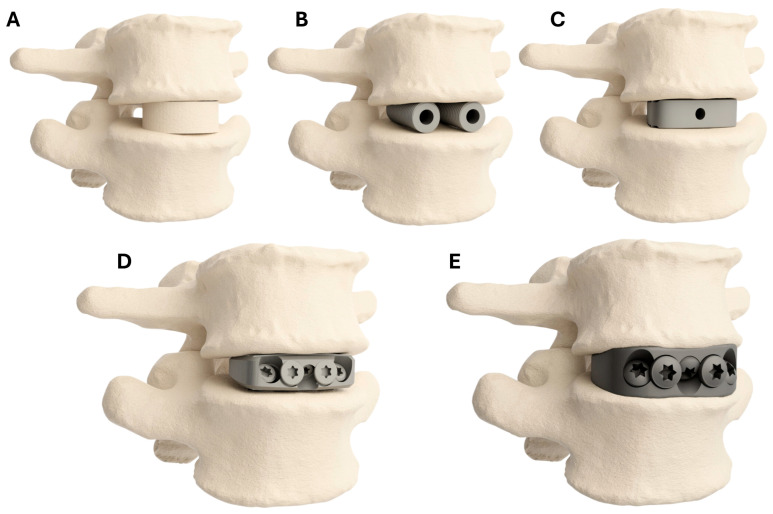
Diagrammatic illustrations of cage design categories: (**A**) Femoral Ring Allograft; (**B**) Bagby and Kuslich style cages; (**C**) non-integral screw fixation cages; (**D**) integral screw fixation cages; (**E**) 3DMorphic Patient-Specific Interbody Cage.

**Figure 5 jpm-15-00320-f005:**
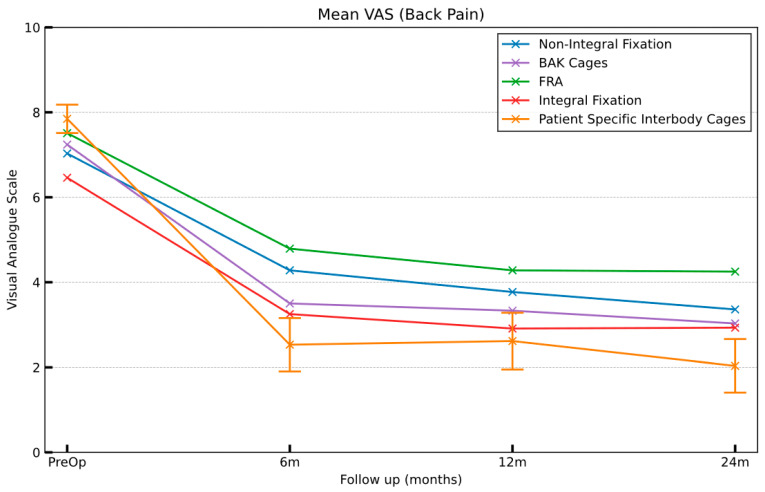
VAS scores for interbody device groups at 6m, 12m, and 24m follow-ups. Error bars represent 95% confidence intervals at each follow-up time point.

**Figure 6 jpm-15-00320-f006:**
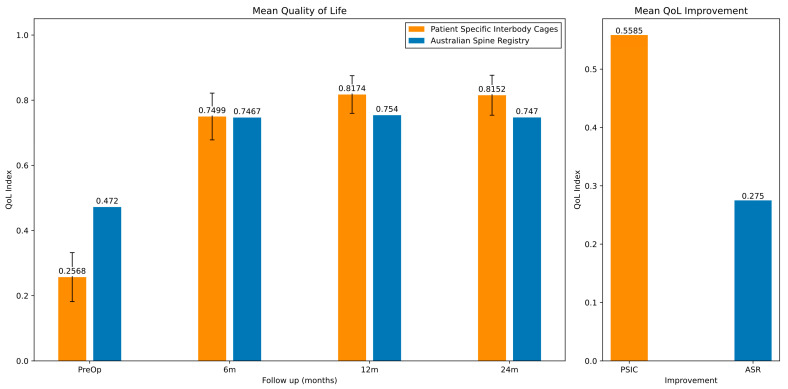
Mean QoL data derived from EQ-5D questionnaires for both the ASR and PSIC cohorts. Error bars represent 95% confidence intervals.

**Figure 7 jpm-15-00320-f007:**
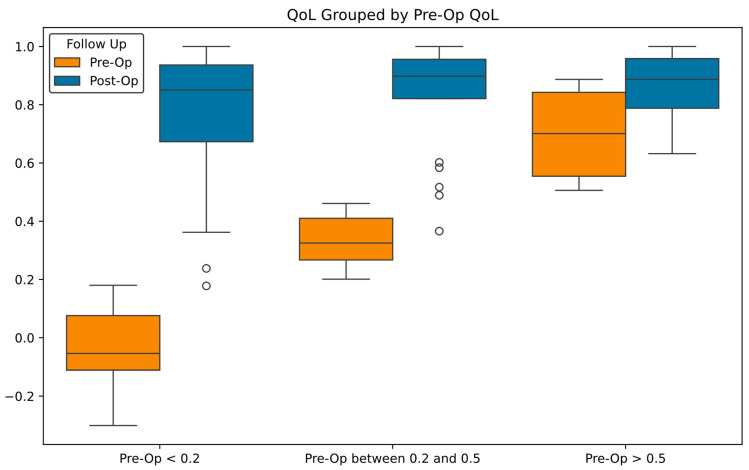
Pre-op and post-op QoL data, grouped by the patient’s pre-op QoL index.

**Figure 8 jpm-15-00320-f008:**
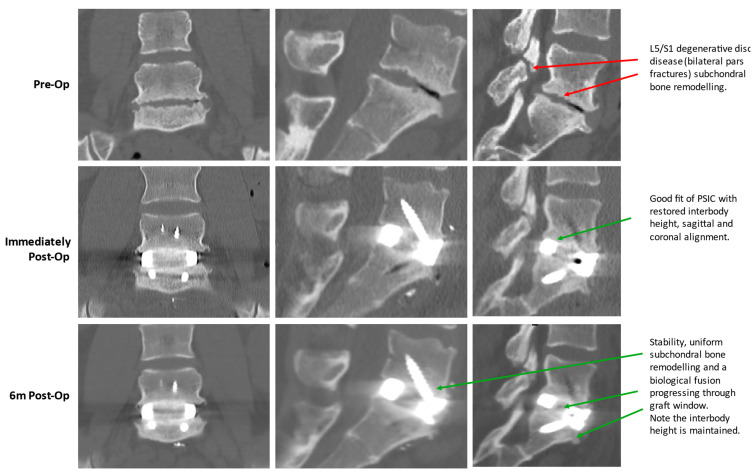
Radiographic assessment of a single-level L5/S1 PSIC ALIF device.

**Table 1 jpm-15-00320-t001:** Patient demographics for the PSIC and OTS cohorts.

	PSIC	OTS—ISF	OTS—NISF	OTS—FRA	OTS—BAK
Sample size (*n*)	78	508	242	274	344
Age (mean)	57.7	52.9	55.6	42.6	42.1
% Male/female	56%	43%	38%	42%	46%
% Circumferential fusion	59%	24%	95%	100%	0%
No. of operated levels (mean)	1.4	1.2	1.2	1.6	1.1

**Table 2 jpm-15-00320-t002:** Risk of bias summary from the 14 RCTs.

	Random Sequence Generation (Selection Bias)	Allocation Concealment (Selection Bias)	Blinding of Participants and Personnel (Performance Bias)	Blinding of Outcome Assessment (Detection Bias)	Incomplete Outcome Data (Attrition Bias)	Selective Reporting (Reporting Bias)	No Other Bias
McKenna et al. [59]							
Chatterjee et al. [56]							
Thalgott et al. [58]							
Blumenthal et al. [57]							
Hoff et al. [53]							
Menezes et al. [55]							
Christensen et al. [52]							
Rickert et al. [54]							
Gornet et al. [51]							
Lavelle et al. [47]							
Delamarter et al. [49]							
Ziegler et al. [48]							
Ohtori et al. [50]							
Butterman et al. [46]							

Legends: 

 = Low risk, 

 = Unclear/some risk, 

 = High risk.

**Table 3 jpm-15-00320-t003:** Summary of study characteristics from the 22 eligible studies identified through the systematic review.

Author	Study Design	Year	Cohort Characteristics	Intervention/Comparator Arm	Mean (Age)	Sample Size	Follow-Up
McKenna et al. [59]	RCT	2005	DDD between L3 and S1 with pain/functional deficit for 6 months and failure to respond to conservative measures for at least 3 months.	Circumferential (ALIF) with SynCage	41.4 (29–65) *	41	>24 months
				Circumferential (ALIF) with FRA	39 (24–53) *	37	
Chatterjee et al. [56]	RCT	2020	Patients were treated for one or more of the following diagnoses: painful DDD, postlaminectomy syndrome, spondylolisthesis, stenosis, failed discectomy, and recurrent disc herniation.	ALIF (bone marrow aspirate graft)	58.95 (14.6) **	20	12 months
				ALIF (cancellous homologous bone chips)	58.33 (33.99) **	21	
Thalgott et al. [58]	RCT	2009	Primary diagnosis of internal disc disruption diagnosed via discography, DDD, or herniated nucleus pulposus at 1 or 2 levels between L3 and S1.	ALIF (freeze dried femoral ring allograft)	44.8 (6.87) **	19	24 months
				ALIF (frozen allograft)	42.29 (10.14) **	21	
Blumenthal et al. [57]	RCT	2005	Symptomatic single-level DDD confirmed by discography, back pain, and/or leg pain with no nerve root compression.	ALIF	39.6 (20–60) *	99	24 months
				Total Disc Replacement (TDR)	39.6 (19–60) *	205	
Hoff et al. [53]	RCT	2015	Two-level DDD with severe low back pain with or without non-radicular leg pain.	Hybrid (ALIF))	45.4 (38–57) *	26	>24 months
				Transforaminal LIF (TLIF	47.6 (36–61) *	24	
Menezes et al. [55]	RCT	2022	Degenerative conditions of the spine.	Extreme LIF (XLIF)	57.04 (3.8) **	30	24 months
						15	
Christensen et al. [52]	RCT	2002	Severe chronic low back/leg pain, static or dynamic, resulting from localized lumbar or lumbosacral segmental instability caused by isthmic spondylolisthesis (grades 1 and 2), primary degeneration (no previous surgery), secondary degeneration after decompressive surgery, or accelerating degeneration after decompressive surgery.	Circumferential (ALIF)	45.4 (20–63) *	75	24 months
				Posterolateral fusion	45.5 (23–65) *	73	
Rickert et al. [54]	RCT	2019	Chronic low back pain, radiculopathy, and spinal claudication symptoms caused by degenerative changes in the lumbar spine.	ALIF (NH-SiO2 Nanobone putty)	60.6 (12.5) **	20	12 months
				ALIF (Homologous bone graft)	66.1 (9.6) **	20	
Gornet et al. [51]	RCT	2011	DDD symptomatic of chronic low back pain.	ALIF	40.2 (18–65)	172	24 months
				TDR	39.9 (18–70)	405	
Lavelle et al. [47]	RCT	2014	Primary diagnosis of DDD at one or two contiguous levels in patients between 21 and 65 years of age; and chronic, disabling low back pain.	ALIF (Stabilis stand alone cage)	43 (NR)	41	24 months
				ALIF (anterior threaded BAK cage)	45.6 (NR)	32	
Delamarter et al. [49]	RCT	2011	DDD at two contiguous vertebral levels from L3 to S1 with or without leg pain.	Circumferential (ALIF)	41.8 (7.81) **	72	24 months
				TDR	41.8 (7.73) **	165	
Ziegler et al. [48]	RCT	2007	Single-level DDD at L3-S1 diagnosed by (i) back and/or leg pain AND (ii) radiographic confirmation.	Circumferential (ALIF)	40.4 (7.6) **	75	24 months
				TDR	38.7 (8) **	161	
Ohtori et al. [50]	RCT	2009	Patients had low back pain continuing for at least 3 years.	ALIF (Discography)	33 (18–43) *	15	>24 months
				ALIF (Discoblock)	36 (20–46) *	15	
Butterman et al. [46]	RCT	2015	Two-level DDD with axial back pain greater than leg symptoms.	Circumferential (ALIF) with posterior open midline approach	45.8 (12.6) **	25	>24 months
				Circumferential (ALIF) with posterior open paraspinal muscle splitting approach	44 (10.6) **	25	
Guyer et al. [60]	Cohort Study	2023	Persistent symptomatic disc degeneration.	ALIF	47.8 (20–77) *	58	>24 months
Hosseini et al. [65]	Cohort Study	2017	Adult Spinal Deformity	ALIF	66.1 (34–85) *	39	12 months
Kashlan et al. [62]	Cohort Study	2020	Degenerative lumbar disease	ALIF	51.9	23	>9 months
Jacob et al. [61]	Cohort Study	2022	DDD	ALIF	51.3 (13.3) **	59	12 months
				TLIF	46.7 (11.0) **	346	
Kuang et al. [63]	Cohort Study	2017	Back or leg pain unresponsive to conservative treatment, noncalcified disc herniation compressing neural structures as confirmed by imaging.	ALIF	52.9 (8.6) **	42	24 months
				TLIF	53.6 (7.4) **	40	
Rao et al. [27]	Cohort Study	2014	ALIF for multiple indications including DDD, Spondylolisthesis, Failed Posterior Fusion, adjacent segment disease, and scoliosis.	ALIF	57 (25–86) *	125	>24 months
Siepe et al. [66]	Cohort Study	2014	Patients were treated for predominant (>80%) low back pain resulting from degenerative disc disease at the L5/S1 level.	ALIF	47 (18.7–73.9) *	71	24 months
Allain et al. [64]	Cohort Study	2014	Back/leg pain unresponsive to conservative treatment, requiring 1 level surgery for degenerative disc disease	ALIF	57.1 (11.1) **	65	12 months

* indicates range, ** indicates SD.

**Table 4 jpm-15-00320-t004:** Summary of raw VAS results from the systematic review.

Author	Cage Information	Cage Subgroup Reference ^1^	Pre- Operative	6M	12M	24M
McKenna et al. [59]	SynCage (Synthes)	NISF-SC	7.1	5.8	6.4	6
	FRA	FRA	7.2	5	4.8	5.2
Chatterjee et al. [56]	4WEB Anterior Spine Truss System	NISF-4W	6.85	3.88	4.05	NR
			7.04	4.14	3.5	NR
Thalgott et al. [58]	FRA	FRA	8.2	6.7	6.2	6
			8	6	4.8	4.9
Blumenthal et al. [57]	BAK Fusion Cage (Zimmer Spine, Minneapolis, MN, USA)	BAK	7.18	4.39	4.04	3.75
	TDR (CHARITE Artificial Disc)	NA	7.2	3.31	3.29	3.12
Hoff et al. [53]	ALIF (PEEK Cage—SynFix LR, Depuy-Synthes); TDR (Maverick, Medtronic, Brampton, Canada)	ISF-SF	6.8	NR	2.7	2.5
	TLIF (PEEK Cage—Capstone, Medtronic)	NA	6.5	NR	3.1	3.8
Menezes et al. [55]	XLIF (PEEK, Nuvasive, San Diego, CA, USA)	NISF-XLIF	7.3	NR	NR	1.6
			6.1	NR	NR	1.5
Christensen et al. [52]	ALIF (Radiolucent Brantigan Cage)	NISF	7	NR	3	3
	PLF (Cotrel–Dubousset Instrumentation)	NA	7	NR	5	5
Rickert et al. [54]	Carbon Fibre-Reinforced Polymer ALIF Cage (DePuy Synthes)	NISF-CF	7.5	3	2	NR
			7	3	3	NR
Gornet et al. [51]	Tapered Fusion Cages (Lumbar Tapered Fusion Device, Medtronic)	BAK	7.33	2.41	2.47	2.36
	Maverick Cobalt Chrome Alloy (Medtronic)	NA	7.17	1.81	1.76	1.8
Lavelle et al. [47]	Stabilis Stand-Alone Cage	BAK	6.9	4.9	4.4	3.6
	Anterior Threaded BAK cages	BAK	7.4	4.8	4.4	3.7
Delamarter et al. [49]	FRA with Posterolateral Fusion	FRA	7.47	4.43	4.03	3.84
	ProDisc-L (Synthes USA)	NA	7.57	3.79	3.56	3.19
Ziegler et al. [48]	FRA with Posterolateral Fusion	FRA	7.5	4.2	4.2	4.3
	ProDisc-L (Synthes, Warsaw, IN, USA)	NA	7.6	4	3.9	3.7
Ohtori et al. [50]	Iliac Bone. No Instrumentation	NA	7	NR	3.2	3.6
			6.5	NR	1	1.6
Butterman et al. [46]	Structural Femoral Cortical Ring Allograft	FRA	7.4	NR	3.4	3.5
			7.3	NR	3.5	2.8
Guyer et al. [60]	STALIF Midline, STALIF M, and STALIF M-Ti Interbody Cage (Centinel Spine)	ISF	6	NR	NR	2.5
Hosseini et al. [65]	ALIF ACR (Nuvasive)	ISF	6.8	NR	2.1	NR
Kashlan et al. [62]	PEEK Cages (Globus, St. Wendel, Germany)	ISF	5.8	3	NR	NR
Jacob et al. [61]	Device not reported. Integral fixation screws used.	ISF	5.8	3.5	2.6	NR
	TLIF	NA	6.5	3.3	3.3	3.6
Kuang et al. [63]	ROI-A Oblique MO-ALIF devices (Zimmer Biomet)	ISF	6.4	2.4	2.5	2.3
	TLIF	NA	6.7	3	2.7	2.6
Rao et al. [27]	Device not reported. Integral fixation screws used.	ISF	7.31	NR	NR	2.7
Siepe et al. [66]	SynFix LR Device used (DePuy Synthes, Warsaw, IN, USA)	ISF	7.52	3.85	4.2	3.9
Allain et al. [64]	ROI-A cage with self-locking plates (Zimmer Biomet, Warsaw, IN, USA)	ISF	5.9	3	2.6	NR

^1^ Integral screw fixation (ISF) and non-integral screw fixation (NISF). See Appendix A for cage reference codes.

**Table 5 jpm-15-00320-t005:** Mean VAS (back pain) scores derived from the systematic review and the PSIC cohort. Statistical significance was calculated against pre-operative baseline * (*p* < 0.05). Note: *n* = number of PSIC patients at each time point.

Mean VAS	Pre-Operative (SD)	6 Months (SD)	12 Months (SD)	24 Months (SD)	Improvement
Non-Integral Screw Fixation	7.03	4.28 *	3.77 *	3.36 *	3.67
BAK Cages	7.24	3.5 *	3.33 *	3.03 *	4.21
FRA	7.51	4.79 *	4.28 *	4.25 *	3.26
Integral Screw Fixation	6.46	3.25 *	2.91 *	2.93 *	3.53
**3DMorphic PSIC**	7.85 (1.50) ***n*** = 78	2.53 (2.26) * ***n*** = 50	2.62 (2.12) * ***n*** = 39	2.03 (2.13) * ***n*** = 44	5.82

**Table 6 jpm-15-00320-t006:** Mean QoL data derived from EQ-5D questionnaires for both the ASR and PSIC cohorts. Note: *n* = number of PSIC patients at each time point.

Mean QoL	Pre-Operative (SD)	6 Months (SD)	12 Months (SD)	24 Months (SD)	Improvement
Australian Spine Registry	0.472	0.747	0.754	0.747	0.275
**3DMorphic PSIC**	0.257 (SD 0.332) ***n*** = 78	0.750 (0.243) ***n*** = 44	0.817 (0.182) ***n*** = 38	0.815 (0.208) ***n*** = 44	0.558

## Data Availability

The original contributions presented in this study are included in the article/Appendix A. Further inquires can be directed to the corresponding author.

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
