# Peer review of "Clinical Outcomes of 3D-Printed Titanium Patient-Specific Implants in Lumbar Interbody Fusion: A Prospective Clinical Trial with a Systematic Review of Conventional Techniques"

_jpm, 2025, doi:10.3390/jpm15070320_

Round 1

Reviewer 1 Report

Comments and Suggestions for Authors

The authors present an engaging and relevant study comparing patient-reported outcome measures (PROMs) in individuals undergoing Lumbar Interbody Fusion (LIF) using 3D-Printed Patient-Specific Interbody Cages (PSIC) versus standard off-the-shelf (OTS) devices. The outcomes for the OTS group were derived through a systematic review of the literature. I would like to offer several suggestions that may help strengthen the quality and clarity of the manuscript:

  • Please specify the exact timeframe for patient recruitment (e.g., January 2022 to December 2024), rather than just the years.
  • Kindly review the use of acronyms throughout the manuscript. Several are repeated unnecessarily (e.g., health-related Quality of Life (QoL) is defined multiple times).
  • Could the authors clarify what type of brace was used for the six weeks following surgery? And why?
  • Where is the risk of bias analysis of the systematic review? I don’t see it in the results. Please include.
  • Please include at least 1 table regarding the preoperative and overall characteristics of the PSIC cohort (age, sex, number of fused levels per procedure...). Then, compare the mean age and number of fused levels of the PSIC group to the OTS devices cohorts to see if they are actually homogeneous and comparable.
  • The authors concluded that devices with additional screw fixation were associated with superior clinical outcomes. However, it would be valuable to elaborate on the outcomes for patients who underwent circumferential fusion—did they also experience improved results?
  • I appreciate the authors’ acknowledgment of a key limitation: that the PSIC group comprised a single patient population, whereas the OTS cohorts were drawn from multiple sources. This certainly impacts the statistical power and generalizability of the findings.
  • While it is true that patient-specific implants are increasingly seen as the future of personalized surgery, the manuscript would benefit from a discussion of associated costs. Why were cost considerations for PSICs not included in the current analysis? Similarly, were cost data available for OTS implants in the reviewed literature? Although PROMs are central to evaluating outcomes, financial sustainability is also a growing concern in modern healthcare systems. Addressing this aspect would help make the article more balanced and transparent, and reduce the perception of bias, even if conflicts of interest have already been disclosed.

In conclusion, this manuscript is not without its flaws. 

Author Response

Thank you for your comments and review.

Please note that page and line numbers were taken from the revised manuscript with tracked changes.

“Please specify the exact timeframe for patient recruitment (e.g., January 2022 to December 2024), rather than just the years.”

  • Recruitment was from February 2022 to July 2024. This has been updated in the manuscript on page 5, line 149.

“Kindly review the use of acronyms throughout the manuscript. Several are repeated unnecessarily (e.g., health-related Quality of Life (QoL) is defined multiple times).”

  • Thank you. This has been updated in the manuscript. PSICs were all changed to PSIC. QoL duplicate removed. Degenerative Disc Disease added. RCTs changed to RCT. Defined BAK. Deleted unnecessary instances of ‘Total Disc Replacement’. Defined NSIF. Updated ISF definition to ‘integral screw fixation cages’. Updated reference to standard deviation (SD).

“Could the authors clarify what type of brace was used for the six weeks following surgery? And why?”

  • Yes, we have updated this in the manuscript to the following, “Patients were braced with a lumbar corset to serve as a psychological reminder to maintain a neutral lumbar spine position and avoid bending, lifting and twisting for six weeks.” Page 8, Lines 214 - 216

“Where is the risk of bias analysis of the systematic review? I don’t see it in the results. Please include.”

  • The risk of bias was initially included in the supplementary information. We have moved this to the results section of the main manuscript. Page 18 Table 4.

“Please include at least 1 table regarding the preoperative and overall characteristics of the PSIC cohort (age, sex, number of fused levels per procedure...). Then, compare the mean age and number of fused levels of the PSIC group to the OTS devices cohorts to see if they are actually homogeneous and comparable.”

  • Individual level patient demographics for the PSIC cohort are included in the supplementary information. We have included a summary table (Page 15, Table 1) that compares the age, sex and other demographics between the PSIC and OTS (lit review) cohort. This is discussed in the discussion section: “The systematic review of conventional fusion techniques provides a comparison for the PSIC cohort. Patients in the PSIC cohort had on average 1.4 levels treated, which was similar across the OTS groups (mean 1.3). 59% of PSIC patients had supplemental posterior instrumentation, compared with 0% of BAK, 24% of ISF, 95% of NISF and 100% of FRA patients. Avoiding supplemental fixation in some patients may help reduce the risks associated with the additional posterior instrumentation [70]. On the other hand, surgeons are increasingly focused on restoring sagittal balance through a circumferential technique [71].” Page 22-23 Lines 436-441.

“The authors concluded that devices with additional screw fixation were associated with superior clinical outcomes. However, it would be valuable to elaborate on the outcomes for patients who underwent circumferential fusion—did they also experience improved results?”

  • We agree. Both FRA and NISF cohorts, with high rates of circumferential fusion compared to other groups, had higher post-operative VAS scores. This indicates that fixation of the interbody cage is a significant contributing factor to VAS scores.
  • We have included the following paragraph in the discussion: “ISF and BAK groups had lower rates of supplemental posterior fixation compared to other groups. NISF and FRA groups had the highest rates circumferential fusion, and were associated with higher post-operative VAS scores. Higher post-operative VAS scores associated with circumferential fusion may reflect an increase in surgical trauma associated with a two-stage front and back circumferential fusion. These results indicate that screw fixation (BAK or ISF) of the interbody cage is a significant contributing factor to lower post-operative VAS scores.” Page 24 Lines 475-480.

“I appreciate the authors’ acknowledgment of a key limitation: that the PSIC group comprised a single patient population, whereas the OTS cohorts were drawn from multiple sources. This certainly impacts the statistical power and generalizability of the findings.”

  • We agree with the reviewer's comments.

“While it is true that patient-specific implants are increasingly seen as the future of personalized surgery, the manuscript would benefit from a discussion of associated costs. Why were cost considerations for PSICs not included in the current analysis? Similarly, were cost data available for OTS implants in the reviewed literature? Although PROMs are central to evaluating outcomes, financial sustainability is also a growing concern in modern healthcare systems. Addressing this aspect would help make the article more balanced and transparent, and reduce the perception of bias, even if conflicts of interest have already been disclosed.”

  • The aim of the present manuscript was to provide clinical outcome data for patients treated with PSIC. We agree that cost-utility and other health economic analyses are needed on PSIC as an emerging technology. However, as the manuscript is already dense with information, such cost analyses were considered beyond the scope of the present study.

We thank the reviewer for their comments. We think that this has improved the quality of the manuscript.

Reviewer 2 Report

Comments and Suggestions for Authors

Dear Authors,

it is my pleasure to review your study.

The article titled "Clinical Outcomes of 3D Printed Titanium Patient-Specific Implants in Lumbar Interbody Fusion; A Prospective Clinical Trial with a Systematic Review of Conventional Techniques" raises an interesting topic but I have a lot of doubts.

1.First, the manuscript should be prepared in accordance with the journal's guidelines.

- the reference format should be improved

- the reference record in the text should be improved

- the abstract should be shortened and divided into appropriate sections. The current abstract looks as if it was prepared for another journal.

2.The purpose of the study should be clearly stated in the abstract and at the end of the introduction. The lack of a study objective significantly complicates its evaluation.

3.Please provide the date of approval from the Bioethics Committee.

4."Patients were recruited from 2022-2024." Please provide the exact date from to.

5.Inclusion and exclusion criteria should be clearly defined.

6."The study was designed as a single region (country), multi-centre, multi-surgeon quantitative prospective cohort study of 78 adult (>18 years) patients undergoing LIF using a 3DP PSIC." It is not entirely clear why the value 78 appears here? This is the section of the methodology that is unclear and incomprehensible.

7.The methodology of the study is full of issues and chaos reigns. It should be remembered that JPM is not a specialist journal of spine surgery. Everything must be clear and obvious to the reader.

In my opinion, the article requires significant correction. Due to many methodological ambiguities, lack of study purpose, and failure to prepare the manuscript in accordance with the journal's guidelines, 

Author Response

Thank you for your review.

Please note that page and line numbers are from the revised manuscript with tracked changes.

“1.First, the manuscript should be prepared in accordance with the journal's guidelines.

- the reference format should be improved

- the reference record in the text should be improved

- the abstract should be shortened and divided into appropriate sections. The current abstract looks as if it was prepared for another journal.”

  • Please see an updated abstract with appropriate background, methods, results and conclusions sections as per journal requirements.
  • Please see updated MDPI reference format.

“2.The purpose of the study should be clearly stated in the abstract and at the end of the introduction. The lack of a study objective significantly complicates its evaluation.”

  • We have updated the abstract and introduction section to include clearly defined aims and purpose of the study: “The purpose of the present study is to contribute data to the field on this new emerging technology of PSIC. The aims of the present study were to investigate health-related Quality of Life (QoL), pain, and complications including revision/reoperation rates, to assess the potential benefits and risks of PSIC. To provide a comparative cohort, we also present a systematic review on patient-reported outcomes of traditional fusion techniques, along with an analysis of the Australian Spine Registry (ASR). The null hypothesis for statistical testing was that there is no difference between PSIC and OTS device clinical scores at any time point.” Page 4 Lines 131-138.

“3.Please provide the date of approval from the Bioethics Committee.”

  • The study was approved by the local HREC on the 6th Feb 2022.
  • This has been updated in the manuscript: “Ethics approval was obtained from the South Eastern Sydney Local Health District (project identifier 2021/ETH11998) on 6 Feb 2022 and the trial was registered with the Australian Therapeutic Goods Administration (CT-2022-CTN-00042-1) on 19 Feb 2022” Page 5 Lines 146-149.

“4.“Patients were recruited from 2022-2024." Please provide the exact date from to.”

  • Recruitment was from February 2022 to July 2024. This has been updated in the manuscript. Page 5 Lines 148-149

We found that the reviewers comments 5, 6 and 7 were best addressed collectively.

“5.Inclusion and exclusion criteria should be clearly defined.”

“6."The study was designed as a single region (country), multi-centre, multi-surgeon quantitative prospective cohort study of 78 adult (>18 years) patients undergoing LIF using a 3DP PSIC." It is not entirely clear why the value 78 appears here? This is the section of the methodology that is unclear and incomprehensible.”

“7.The methodology of the study is full of issues and chaos reigns. It should be remembered that JPM is not a specialist journal of spine surgery. Everything must be clear and obvious to the reader.”

  • Thank you for these comments. We have made significant changes to the structure of the methods section using standard research subheadings. This has been designed in a logical manner from overall study design, ethical approval, power analysis, inclusion/exclusion criteria, PSIC design, operative technique, post-operative care, outcome measures and follow up protocol. We think this improves the structure and clarity of the methods section to ensure the methods are comprehensible and relevant for JPM readership.

Thank you for your comments and review. We think that these have improved the quality of the manuscript.

Round 2

Reviewer 1 Report

Comments and Suggestions for Authors

The authors have addressed my comments. The article is suitable for publication in the present journal.

Author Response

Thank you again for your review.

Reviewer 2 Report

Comments and Suggestions for Authors

Dear Authors,

the changes introduced improve the scientific quality of the manuscript.

However, further correction is necessary.

1.The abstract is still too long. I propose to shorten it a bit.

2."Patients were recruited from February 2022 to July 2024." - please provide a more exact date.

3.The inclusion criteria are too general and still require correction:
-what does "mechanical low back pain" mean?

Based on the criteria presented, will every patient with LBP qualify?

4.The conclusion requires correction:
Excerpt "The results presented in the present study are for PSIC produced by a single company. Therefore, the results presented here may not be applicable to PSIC produced by other companies using different design and manufacturing methods." should be moved to the limitations section.

Author Response

Thank you again for your review.

1.The abstract is still too long. I propose to shorten it a bit.

We have revised the abstract. The length is now 350 words. As the manuscript covers a lot of information, we have tried to reduce the abstract as much as possible whilst ensuring that it covers the key findings of the paper.

2."Patients were recruited from February 2022 to July 2024." - please provide a more exact date.

Patients were recruited from 19 February 2022 to 15 July 2024. This has been updated in the methods.

3.The inclusion criteria are too general and still require correction:
-what does "mechanical low back pain" mean?

Based on the criteria presented, will every patient with LBP qualify?

Thank you for highlighting this. Only patients with discogenic and/or mechanical low back pain that was symptomatic of degenerative disc disease were included. This has been updated in the methods and abstract.

“All patients were selected for surgery due to discogenic and/or mechanical low back pain symptomatic of Degenerative Disc Disease (DDD).” Page 4 Lines 125-126.

4.The conclusion requires correction:
Excerpt "The results presented in the present study are for PSIC produced by a single company. Therefore, the results presented here may not be applicable to PSIC produced by other companies using different design and manufacturing methods." should be moved to the limitations section.

We have moved this to the limitations section.